# TRANSFORMATION-BASED MODELS OF VIDEO SEQUENCES

**Joost van Amersfoort,**[*] **Anitha Kannan, Marc'Aurelio Ranzato,**
**Arthur Szlam, Du Tran & Soumith Chintala**
Facebook AI Research
`joost@joo.st, {akannan, ranzato, aszlam, trandu, soumith}@fb.com`

## ABSTRACT

In this work we propose a simple unsupervised approach for next frame prediction in video. Instead of directly predicting the pixels in a frame given past frames, we predict the transformations needed for generating the next frame in a sequence, given the transformations of the past frames. This leads to sharper results, while using a smaller prediction model.

In order to enable a fair comparison between different video frame prediction models, we also propose a new evaluation protocol. We use generated frames as input to a classifier trained with ground truth sequences. This criterion guarantees that models scoring high are those producing sequences which preserve discriminative features, as opposed to merely penalizing any deviation, plausible or not, from the ground truth. Our proposed approach compares favourably against more sophisticated ones on the UCF-101 data set, while also being more efficient in terms of the number of parameters and computational cost.

## 1 INTRODUCTION

There has been an increased interest in unsupervised learning of representations from video sequences (Mathieu et al., 2016; Srivastava et al., 2015; Vondrick et al., 2016). A popular formulation of the task is to learn to predict a small number of future frames given the previous K frames; the motivation being that predicting future frames requires understanding how objects interact and what plausible sequences of motion are. These methods directly aim to predict pixel values, with either MSE loss or adversarial loss.

In this paper, we take a different approach to the problem of next frame prediction. In particular, our model operates in the space of transformations between frames, directly modeling the source of variability. We exploit the assumption that the transformations of objects from frame to frame should be smooth, even when the pixel values are not. Instead of predicting pixel values, we directly predict how objects transform. The key insight is that while there are many possible outputs, predicting one such transformation will yield motion that may not correspond to ground truth, yet will be realistic; see fig. 1. We therefore propose a *transformation-based* model that operates in the space of affine transforms. Given the affine transforms of a few previous frames, the model learns to predict the local affine transforms that can be deterministically applied on the image patches of the previous frame to generate the next frame. The intuition is that estimation errors will lead to a slightly different yet plausible motion. Note that this allows us to keep using the MSE criterion, which is easy to optimize, as long as it is in transformation space. No blur in the pixel space will be introduced since the output of the transformation model is directly applied to the pixels, keeping sharp edges intact. Refer to fig. 5 and our online material [1] for examples.

The other contribution of this work is the evaluation protocol. Typically, generative models of video sequences are evaluated in terms of MSE in pixel space (Srivastava et al., 2015), which is not a good choice since this metric favors blurry predictions over other more realistic looking options that just happen to differ from the ground truth. Instead, we propose to feed the generated frames to a video

---

[*]Work done as part of internship with FAIR
[1]see: `http://joo.st/ICLR/GenerationBenchmark`

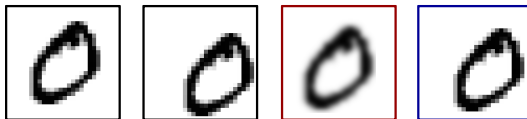

Figure 1: Motivating toy example. From left to right: the first digit shows what the model is conditioned upon, the second digit shows the frame we would like to predict at the next time step, the third digit shows the blurry prediction if we were to minimize MSE in pixel space, the last digit shows the prediction when minimizing MSE in the space of transformations. While the two models may have the same MSE in pixel space, the transformation-based model generates much sharper outputs. Although the motion is different than the ground truth (second digit), it is still a plausible next frame to the conditioned frame. In practice, the input is a sequence of consecutive frames.

classifier trained on ground truth sequences. The idea is that the less the classifier's performance is affected by the generates frames the more the model has preserved distinctive features and the more the generated sequences are plausible. Regardless of whether they resemble the actual ground truth or not. This protocol treats the classifier as a black box to measure how well the generated sequences can serve as surrogate for the truth sequence for the classification task. In this paper we will validate our assumption that motion can be modelled by local affine transforms, after which we will compare our method with networks trained using adversarial training and simple regression on the output frame, using both this new evaluation protocol and by providing samples for qualitative inspection.

Our experiments show that our simple and efficient model outperforms other baselines, including much more sophisticated models, on benchmarks on the UCF-101 data set (Soomro et al., 2012). We also provide qualitative comparisons to the moving MNIST digit data set (Srivastava et al., 2015).

## 1.1 RELATED WORK

Early work on video modeling focused on predicting small patches (Michalski et al., 2014; Srivastava et al., 2015); unfortunately, these models have not shown to scale to the complexity of high-resolution videos. Also these models require a significant amount of parameters and computational power for even relatively simple data.

In Ranzato et al. (2014), the authors circumvented this problem by quantizing the space of image patches. While they were able to predict a few high-resolution frames in the future, it seems dissatisfying to impose such a drastic assumption to simplify the prediction task.

Mathieu et al. (2016) recently proposed to replace MSE in pixel space with a MSE on image gradients, leveraging prior domain knowledge, and further improved using a multi-scale architecture with adversarial training (Goodfellow et al., 2014). While producing better results than earlier methods, the models used require a very large amount of computational power. We make an explicit comparison to this paper in the experiments section 3.

In Oh et al. (2015), frames of a video game are predicted given an action (transformation) taken by the player. While the paper shows great results, the movement in a natural video cannot be described by a simple action and is therefore not widely applicable. Finally, our work is also related to optical flow estimation (Brox et al., 2004). Instead of estimating the flow of pixels, here we estimate the flow of patches and separately predict how these patches transform in future frames.

Prior work relating to the evaluation protocol can be found in Yan et al. (2015). The authors generate images using a set of predefined attributes and later show that they can recover these using a pretrained neural network. Our proposal extends this to videos, which is more complicated since both appearance and motion are needed for correct classification.

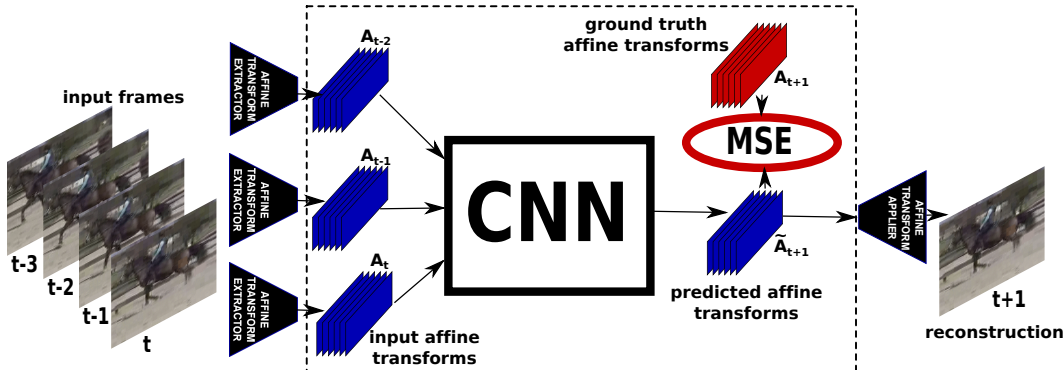

Figure 2: Outline of the transformation-based model. The model is a CNN that takes as input a sequence of consecutive affine transforms between pairs of adjacent video frames. It predicts the affine transform between the last input frame and the next one in the sequence. We compute affine transforms (6 parameters per patch) for overlapping patches of size $8 \times 8$ in each video frame. Learning operates in the space of transformations as shown inside the dashed box. The front-end on the left is a module that estimates the affine transforms between pairs of consecutive input frames. The post-processor on the right reconstructs a frame from the predicted set of affine transforms and it is only used at test time.

## 2   MODEL

The model we propose is based on three key assumptions: 1) just estimating object motion yields sequences that are plausible and relatively sharp, 2) global motion can be estimated by tiling high-resolution video frames into patches and estimating motion "convolutionally" at the patch level, and 3) patches at the same spatial location over two consecutive time steps undergo a deformation which can be well described by an affine transformation.

The first assumption is at the core of the proposed method: by considering uncertainty in the space of transformations we produce sequences that may still look plausible. The other two assumptions state that a video sequence can be composed by patches undergoing affine transformations. We agree that these are simplistic assumptions, which ignore how object identity affects motion and do not account for out of plane rotations and more general forms of deformation. However, our qualitative and quantitative evaluation shows the efficacy of these assumptions to real video sequence as can be seen in section 3 and from visualizations in the supplementary material[2].

Our approach consists of three steps. First, we estimate affine transforms of every video sequence to build a training set for our model. Second, we train a model that takes the past $N$ affine transforms and predicts the next $M$ affine transforms. Finally, at test time, the model uses the predicted affine transforms to reconstruct pixel values of the generated sequence. We describe the details of each phase in the following sections.

### 2.1   AFFINE TRANSFORM EXTRACTOR

Given a frame $x$ and the subsequent frame $y$, the goal of the affine transform extractor is to learn mappings that can warp $x$ into $y$. Since different parts of the scene may undergo different transforms, we tile $x$ into overlapping patches and infer a transformation for each patch. The estimation process couples the transformations at different spatial locations because we minimize the reconstruction error of the entire frame $y$, as opposed to treating each patch independently.

---

[2]see: http://joo.st/ICLR/ReconstructionsFromGroundTruth and http://joo.st/ICLR/GenerationBenchmark

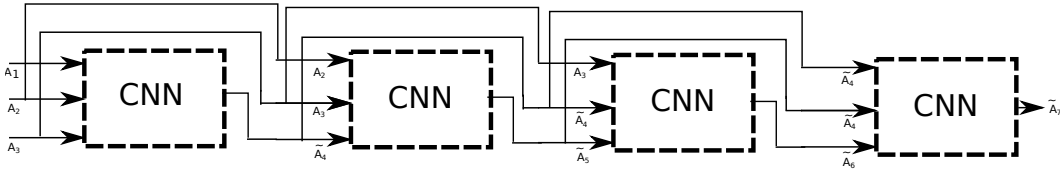

Figure 3: Outline of the system predicting 4 frames ahead in time. Only affine transforms $A_1$, $A_2$ and $A_3$ are provided, and the model predicts $\tilde{A}_4$, $\tilde{A}_5$, $\tilde{A}_6$ and $\tilde{A}_7$, which are used to reconstruct the next 4 frames. Since affine parameters are continuous values and the whole chain of CNNs is differentiable, the whole unrolled system can be trained by back-propagation of the error. Note that CNNs all share the same parameters

Let $x$ and $y$ have size $D_r \times D_c$. Let image $x$ be decomposed into a set of overlapping patches, each containing pixels from patches of size $d_r \times d_c$ with $d_r \le D_r$ and $d_c \le D_c$. These patches are laid out on a regular grid with stride $s_r$ and $s_c$ pixels over rows and columns, respectively. Therefore, every pixel participates in $\frac{d_r}{s_r}\frac{d_c}{s_c}$ overlapping patches, not taking into account for the sake of simplicity border effects and non-integer divisions. We denote the whole set of overlapping patches by $\{X_k\}$, where index $k$ runs over the whole set of patches. Similarly and using the same coordinate system, we denote by $\{Y_k\}$ the set of overlapping patches of $y$.

We assume that there is an affine mapping $A_k$ that maps $X_k$ to $Y_k$, for all values of $k$. $A_k$ is a $2 \times 3$ matrix of free parameters representing a generic affine transform (translation, rotation and scaling) between the coordinates of output and input frame. Let $\tilde{Y}_k$ be the transformed patches obtained when $A_k$ is applied to $X_k$. Since coordinates overlap between patches, we reconstruct $y$ by averaging all predictions at the same location, yielding the estimate $\tilde{y}$. The joint set of $A_k$ is then jointly determined by minimizing the mean squared reconstruction error between $y$ and $\tilde{y}$.

Notice that our approach and aim differs from spatial transformer networks (Jaderberg et al., 2015) since we perform this estimation off-line only for the input frames, computing one transform per patch.

In our experiments, we extracted $16 \times 16$ pixel patches from the input and we used stride 4 over rows and columns. The input patches are then matched at the output against smaller patches of size $8 \times 8$ pixels, to account for objects moving in and out of the patch region.

## 2.2 Affine Transform Predictor

The affine transform predictor is used to predict the affine transforms between the last input frame and the next frame in the sequence. A schematic illustration of the system is shown in fig. 2. It receives as input the affine transforms between pairs of adjacent frames, as produced by the affine transform extractor described in the previous section. Each transform is arranged in a grid of size $6 \times n \times n$, where $n$ is the number of patches in a row/column and 6 is the number of parameters of each affine transform. Therefore, if four frames are used to initialize the model, the actual input consists of 18 maps of size $n \times n$, which are the concatenation of $A_{t-2}, A_{t-1}, A_t$, where $A_t$ is the collection of patch affine transforms between frame at time $t - 1$ and $t$.

The model consists of a multi-layer convolutional network without any pooling. The network is the composition of convolutional layers with ReLU non-linearity, computing a component-wise thresholding as in $v = \max(0, u)$. We learn the parameters in the filters of the convolutional layers by minimizing the mean squared error between the output of the network and the target transforms.

Notice that we do not add any regularization to the model. In particular, we rely on the convolutional structure of the model to smooth out predictions at nearby spatial locations.

## 2.3 MULTI-STEP PREDICTION

In the previous section, we described how to predict the set of affine transforms at the next time step. In practice, we would like to predict several time steps in the future.

A greedy approach would: a) train as described above to minimize the prediction error for the affine transforms at the next time step, and b) at test time, predict one step ahead and then re-circulate the model prediction back to the input to predict the affine transform two steps ahead, etc. Unfortunately, errors may accumulate throughout this process because the model was never exposed to its own predictions at training time.

The approach we propose replicates the model over time, also during training as shown in fig. 3. If we wish to predict $M$ steps in the future, we replicate the CNN $M$ times and pass the output of the CNN at time step $t$ as input to the same CNN at time step $t + 1$, as we do at test time. Since predictions live in a continuous space, the whole system is differentiable and amenable to standard back-propagation of the error. Since parameters of the CNN are shared across time, the overall system is equivalent to a peculiar recurrent neural network, where affine transforms play the role of recurrent states. The experiments in section 3 demonstrate that this method is more accurate and robust than the greedy approach.

## 2.4 TESTING

At test time, we wish to predict $M$ frames in the future given the past $N$ frames. After extracting the $N - 1$ affine transforms from the frames we condition upon, we replicate the model $M$ times and feed its own prediction back to the input, as explained in the previous section.

Once the affine transforms are predicted, we can reconstruct the actual pixel values. We use the last frame of the sequence and apply the first set of affine transforms to each patch in that frame. Each pixel in the output frame is predicted multiple times, depending on the stride used. We average these predictions and reconstruct the whole frame. As required, we can repeat this process for as many frames as necessary, using the last reconstructed frame and the next affine transform.

In order to evaluate the generation, we propose to feed the generated frames to a trained classifier for a task of interest. For instance, we can condition the generation using frames taken from video clips which have been labeled with the corresponding action. The classifier has been trained on ground truth data but it is evaluated using frames fantasized by the generative model. The performance of the classifier on ground truth data is an upper bound on the performance of any generative model. This evaluation protocol does not penalize any generation that deviates from the ground truth, as standard MSE would. It instead check that discriminative features and the overall semantics of the generated sequence is correct, which is ultimately what we are interested in.

## 3 EXPERIMENTS

In this section, we validate the key assumptions made by our model and compare against state-of-the-art generative models on two data sets. We strongly encourage the reader to watch the short video clips in the Supplementary Material to better understand the quality of our generations.

In section 2, we discussed the three key assumptions at the foundations of our model: 1) errors in the transformation space look still plausible, 2) a frame can be decomposed into patches, and 3) each patch motion is well modeled by an affine transform. The results in the Supplementary Material [3] validate assumption 2 and 3 qualitatively. Every row shows a sequence from the UCF-101 dataset (Soomro et al., 2012). The column on the left shows the original video frames and the one on the right the reconstructions from the estimated affine transforms, as described in section 2.1. As you can see there is barely any noticeable difference between these video sequences, suggesting that video sequences can be very well represented as tiled affine transforms. For a quantitative comparison and for an assessment of how well the first assumption holds, please refer to section 3.2.

---

[3]see: http://joo.st/ICLR/ReconstructionsFromGroundTruth

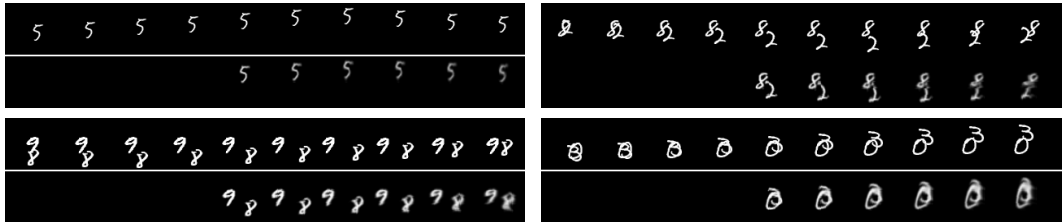

Figure 4: Predictions of 4 sequences from the moving MNIST dataset. The top row of each pair shows the ground truth frames; the first four frames are used as input to the model. The bottom row shows the predictions of the model.

In the next section, we will first report some results using the toy data set of "moving MNIST digits" (Srivastava et al., 2015). We then discuss generations of natural high-resolution videos using the UCF-101 dataset and compare to current state-of-the-art methods.

## 3.1 MOVING MNIST

For our first experiment, we used the dataset of moving MNIST digits (Srivastava et al., 2015) and perform qualitative analysis[4]. It consists of one or two MNIST digits, placed at random locations and moving at constant speed inside a $64 \times 64$ frame. When a digit hits a boundary, it bounces, meaning that velocity in that direction is reversed. Digits can occlude each other and bounce off walls, making the data set challenging.

Using scripts provided by Srivastava et al. (2015), we generated a fixed dataset of 128,000 sequences and used 80% for training, 10% for validation and 10% for testing. Next, we estimated the affine transforms between every pair of adjacent frames to a total of 4 frames, and trained a small CNN in the space of affine transforms. The CNN has 3 convolutional layers and the following number of feature maps: 18, 32, 32, 6. All filters have size $3 \times 3$.

Fig. 4 shows some representative test sequences and the model outputs. Each subfigure corresponds to a sequence from the test set; the top row corresponds to the ground truth sequence while the bottom row shows the generations. The input to the CNN are three sets of affine transforms corresponding to the first four consecutive frames. The network predicts the next six sets of affine transforms from which we reconstruct the corresponding frames. These results should be compared to fig. 5 in Srivastava et al. (2015). The generations in fig. 4 show that the model has potential to represent and generate video sequences, it learns to move digits in the right direction, to bounce them, and it handles multiple digits well except when occluion makes inputs too ambiguous. The model's performance is analyzed quantitatively in the next section using high resolution natural videos.

## 3.2 UCF 101 DATA SET

The UCF-101 dataset (Soomro et al., 2012) is a collection of 13320 videos of 101 action categories. Frames have size $240 \times 320$ pixels. We train a CNN on patches of size $64 \times 64$ pixels; the CNN has 6 convolutional layers and the following number of feature maps: 18, 128, 128, 128, 64, 32, 16, 6. All filters have size $3 \times 3$. The optimal number of filters has been found using cross-validation in order to minimize the estimation error of the affine transform parameters. Unless otherwise stated, we condition generation on 4 ground truth frames and we predict the following 8 frames.

We evaluate several models[5]: a) a baseline which merely copies the last frame used for conditioning, b) a baseline method which estimates optical flow (Brox et al., 2004) from two consecutive frames

---

[4]A quantitative analysis would be difficult for this data set because metrics reported in the literature like MSE (Srivastava et al., 2015) are not appropriate for measuring generation quality, and it would be difficult to use the metric we propose because we do not have labels at the sequence level and the design of a classifier is not trivial.

[5]Unfortunately, we could not compare against the LSTM-based method in Srivastava et al. (2015) because it does not scale to high-resolution videos, but only to small patches.

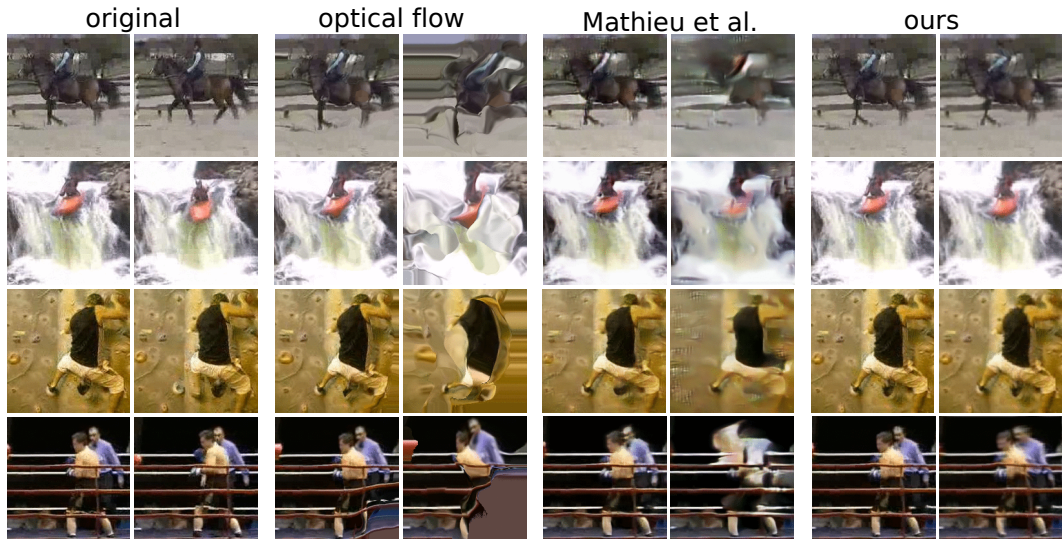

Figure 5: Example of predictions produced by different models. Each row shows an example. The first two columns show the ground truth. The two frames are 4 time steps apart. The next two columns show predictions from a baseline model employing optical flow. Next, we show the prediction produced by the adversarially trained CNN proposed by Mathieu et al. (2016). The last two column show the prediction produced by our affine-transformation based approach. All pairs in the same column group are four time steps apart. All methods were conditioned on the same set of 4 input frames (not shown in the figure)

Table 1: Classification accuracy on UCF-101 dataset. The classifier is trained on the actual training video sequences, but it is tested using frames generated by various generative models. Each column shows the accuracy on the test set when taking a different number of input frames as input. Our approach maps $16 \times 16$ patches into $8 \times 8$ with stride 4, and it takes 4 frames at the input.

| Method | 4 frames | 8 frames |
|---|---|---|
| Ground truth frames | 72.46 | 72.29 |
| Using ground truth affine transforms | 71.7 | 71.28 |
| Copy last frame | 60.76 | 54.27 |
| Optical Flow | 57.29 | 49.37 |
| Mathieu et al. (2016) | 57.98 | 47.01 |
| ours - one step prediction (not unrolled) | 64.13 | 57.63 |
| ours - four step prediction (unrolled 4 times) | 64.54 | 57.88 |

and extrapolates flow in subsequent frames under the assumption of constant flow speed, c) an adversarially trained multi-scale CNN (Mathieu et al., 2016) and several variants of our proposed approach.

Qualitative comparisons can be seen in the fig. 5 and in the supplementary material[6]. The first column on the page shows the input, the second the ground truth, followed by results from our model, Mathieu et al. (2016) and optical flow (Brox et al., 2004). Note especially the severe deformations in the last two columns, while our model keeps the frame recognizable. It produces fairly sharp reconstructions validating our first hypothesis that errors in the space of transformations still yield plausible reconstructions (see section 2). However it is also apparent that our approach underestimates movement, which follows directly from using the MSE criterion. As discussed before, MSE in pixel space leads to blurry results, however using MSE in transformation space also has some drawbacks. In practice, the model will predict the average of several likely transformations, which could lead to an understimation of the true movement.

---

[6]see: http://joo.st/ICLR/GenerationBenchmark

In order to quantify the generation quality we use the metric described in section 2.4. We use C3D network (Tran et al., 2015) as the video action classifier: C3D uses both appearance and temporal information jointly, and is pre-trained with Sports1M (Karpathy et al., 2014) and fine tuned on UCF 101. Due to the model constraints, we trained only two models, that takes 4 and 8 frames as input, respectively.

We evaluate the quality of generation using 4 (the first four predicted frames) and the whole set of 8 predicted frames, for the task of action classification. At test time, we generate frames from each model under consideration, and then use them as input to the corresponding C3D network.

Table 1 shows the accuracy of our approach and several baselines. The best performance is achieved by using ground truth frames, a result comparable to methods recently appeared in the literature (Karpathy et al., 2014; Tran et al., 2015). We see that for ground truth frames, the number of frames (4 or 8) doesn't make a difference. There is not much additional temporal or spatial signal provided by having greater than four frames. Next, we evaluate how much we lose by representing frames as tiled affine transforms. As the second row shows there is negligible if any loss of accuracy when using frames reconstructed from the estimated affine transforms (using the method described in section 2.1), validating our assumptions at the beginning of section 2 on how video sequences can be represented. The next question is then whether these affine transforms are predictable at all. The last two rows of Table 1 show that this is indeed the case, to some extent. The longer the sequence of generated frames the poorer the performance, since the generation task gets more and more difficult.

Compared to other methods, our approach performs better than optical flow and even the more sophisticated multi-scale CNN proposed in Mathieu et al. (2016) while being computationally cheaper. For instance, our method has less than half a million parameters and requires about 2G floating point operations to generate a frame at test time, while the multi-scale CNN of Mathieu et al. (2016) has 25 times more parameters (not counting the discriminator used at training time) and it requires more than 100 times more floating point operations to generate a single frame.

Finally, we investigate the robustness of the system to its hyper-parameters: a) choice of patch size, b) number of input frames, and c) number of predicted frames. The results reported in Table 2 demonstrate that the model is overall pretty robust to these choices. Using patch sizes that are too big makes reconstructions blocky but within each block motion is coherent. Smaller patch sizes give more flexibility but make the prediction task harder as well. Mapping into patches of size smaller than $16 \times 16$ seems a good choice. Using only 2 input frames does not seem to provide enough context to the predictor, but anything above 3 works equally well. Training for prediction of the next frame works well, but better results can be achieved by training to predict several frames in the future, overall when evaluating longer sequences.

Table 2: Analysis of the robustness to the choice of hyper-parameters, shows classification scores compared to reference model. The reference model takes 4 frames as input, predicts one frame, and maps $12 \times 12$ patches onto $8 \times 8$ patches with stride 4.

| Method | 4 frames | 8 frames |
|---|---|---|
| reference | 63.57 | 57.32 |
| Varying patch size | | |
| from $32 \times 32$ to $16 \times 16$ | 61.73 | 53.85 |
| from $16 \times 16$ to $8 \times 8$ | 63.75 | 57.18 |
| Number of input frames | | |
| 2 | 63.6 | 57.11 |
| 3 | 63.8 | 57.4 |
| Number of predicted frames | | |
| 2 | 64.1 | 57.5 |
| 4 | 64.54 | 57.88 |

# 4 CONCLUSIONS

In this work, we proposed a new approach to generative modeling of video sequences. This model does not make any assumption about the spatio-temporal resolution of video sequences nor about object categories. The key insight of our approach is to model in the space of transformations as opposed to raw pixel space. *A priori* we lack a good metric to measure how well a frame is reconstructed under uncertainty due to objects motion in natural scenes. Uncertainty about object motion and occlusions causes blurry generations when using MSE in pixel space. Instead, by operating in the space of transformations we aim at predicting how objects move, and estimation errors only yield a different, and possibly still plausible, motion. With this motivation we proposed a simple CNN operating in the space of affine transforms and we showed that it can generate sensible sequences up to about 4 frames. This model produces sequences that are both visually and quantitatively better than previously proposed approaches.

The second contribution of this work is the metric to compare generative models of video sequences. A good metric should not penalize a generative model for producing a sequence which is plausible but different from the ground truth. With this goal in mind and assuming we have at our disposal labeled sequences, we can first train a classifier using ground truth sequences. Next, the classifier is fed with sequences produced by our generative model for evaluation. A good generative model should produce sequences that still retain discriminative features. In other words, plausibility of generation is assessed in terms of how well inherent information is preserved during generation as opposed to necessarily and merely reproducing the ground truth sequences.

The proposed model is relatively simple; straightforward extensions that could improve its prediction accuracy are the use of a multi-scale architecture and the addition of recurrent units. These would enable a better modeling of objects of different sizes moving at varying speeds and to better capture complex temporal dynamics (e.g., cyclical movements like walking). A larger extension would be the addition of an appearance model, which together with our explicit transformation model could lead to learning better feature representations for classification.

In our view, the proposed approach should be considered as a stronger baseline for future research into next frame prediction. Even though our analysis shows improved performance and better looking generations, there are also obvious limitations. The first such limitation is the underestimation of transformations due to usage of the MSE as a criterion. We consider two main avenues worth pursuing in this space. First, we consider modelling a distribution of transformations and sampling one from it. The challenge of this approach is to sample a consistent trajectory. One could model the distribution of an entire trajectory, but that is a complex optimization problem. A second option is to use adversarial training to force the model to pick a plausible action. This option does not guarantee that underestimation of movement will be avoided. This will depend on the discriminator model accepting this as a plausible option.

Another limitation is that the current model does not factor out the "what" from the "where", appearance from motion. The representation of two distinct objects subject to the same motion, as well as the representation of the same object subject to two different motion patterns are intrinsically different. Instead, it would be more powerful to learn models that can discover such factorization and leverage it to produce more efficient and compact representations.

ACKNOWLEDGMENTS

Authors thank Camille Couprie and Michael Mathieu for discussions and helping with evaluation of their models.

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
