# Peer review of "Transformation-based Models of Video Sequences"

_ICLR 2017 — rejected_

[Public Comment · (anonymous) · 13 Nov 2016]
**Missing relevant reference and comparison?**

The transformation-based approach for generating the next frame in a sequence was used in ICLRw2016

[Public Comment · ICLR 2017 conference · 05 Dec 2016]
**Question regarding related work**

Could the authors comment on the relationship between their work and these previous works that appear to use a similar transformation-based video prediction technique?

Dynamic Filter Networks (NIPS 2016)
Unsupervised Learning for Physical Interaction through Video Prediction (NIPS 2016)
Visual Dynamics: Probabilistic Future Frame Synthesis via Cross Convolutional Networks (NIPS 2016)

[Official Review · AnonReviewer1 · rating 3 · confidence 3 · 14 Dec 2016 (modified: 16 Dec 2016)]
**An interesting direction, but has many flaws**

This paper describes an approach to predict (unseen) future frames of a video given a set of known past frames. The approach is based on a CNN that, in contrast to most related papers, work in the space of affine transformations (instead of pixels or flow). Said another way, the network takes as input a set of affine transforms that describe the motion of patches in the past frames, and likewise, outputs a set of affine transforms that predict future patch motion.

To that aim, the authors make a few simplifying hypotheses, namely, that a sequence of frames can be modeled accurately enough in their patch-affine framework. This is not unreasonable. A lot of papers in the optical flow community are based on similar hypotheses, i.e. model the flow as a smoothly varying affine field (for instance see "Locally affine sparse-to-dense matching for motion and occlusion estimation" by Leordeanu et al., "EpicFlow: Edge-Preserving Interpolation of Correspondences for Optical Flow" by Revaud et al., "Optical Flow With Semantic Segmentation and Localized Layers" by Sevilla-Lara et al.). These methods are state of the art, which gives a hint about the validity of this kind of approach. 
In addition, it also seems very reasonable to reformulate the prediction task as predicting motion rather than predicting raw pixels. Indeed, the (patch-affine) motion space is considerably smaller than the image space, making the problem much more tractable and amenable to high-resolution videos.

While I agree with the authors on these points, I also find that the paper suffer from important flaws. Specifically:

  - the choice of not comparing with previous approaches in term of pixel prediction error seems very "convenient", to say the least. While it is clear that the evaluation metric is imperfect, it is not a reason to completely dismiss all quantitative comparisons with previous work. The frames output by the network on, e.g. the moving digits datasets (Figure 4), looks ok and can definitely be compared with other papers. Yet, the authors chose not to, which is suspicious.
  
  - The newly proposed metric poses several problems. First, action classification is evaluated with C3D, which is not a state-of-the-art approach at all for this task. Second, this metric actually *does not* evaluate what the network is claimed to do, that is, next frame prediction. Instead, it evaluates if another network, which was never trained to distinguish between real or synthetic frames by the way, can accurately classify an action from the predicted frames. I find that this proxy metric is only weakly related to what is supposed to be measured. In adition, it does not really make sense to train a network for something else that the final task it is evaluated for.
  
  - how is the affine motion of patches estimated? It is only explained that the problem is solved globally (not treating each patch independently) in a pretty vague manner. Estimating the motion of all patches is akin to solving the optical flow, which is still an active subject of research. Therefore, an important flaw of the paper lies in the potentially erroneous etimation of the motion input to the network. In the videos made available, it is clear that the motion is wrongly estimated sometimes. Since the entire approach depends on this input, I find it important to discuss this aspect. How do motion estimation failures impact the network? Also, the patch-affine hypothesis does not hold when patches are large enough that they cover several objects with contradictory motion. Which appears to be the case on UCF101 videos.
  
  - Even ignoring the weird proxy-evaluation part, the network is still not trained end-to-end. That is, the network is trained to minimize the difference between (noisy) ground-truth and output affine transforms, instead of minimizing a loss in the actual output space (frame pixels) for which an (exact) ground-truth is available. It is true that the MSE loss on raw pixels leads to blurry results, but other types of losses do exist, for instance the gradient loss introduced by Mathieu et al. was shown to solve this issue. As noted by the authors themselves, minimizing a loss in the transformation space, where affine parameters are harder to intepret, introduces unexpected artifacts. The motion is often largely underestimated, as is obvious in Figure 5 where it is hard to tell the difference between the input and output frames. 
  
  - The proposed approach is not sufficiently compared to previous work. In particular, the approach is closely related to "SPATIO-TEMPORAL VIDEO AUTOENCODER WITH DIFFERENTIABLE MEMORY" of Taraucean et al, ICLR'15. This paper also output prediction in the motion space. Experimental results should compare against it.

  - The comparison with optical flow is unfair. First, the approach of Brox et al. is more than 10 years old. Second, it is not really fair to assume a constant flow for all frames. At least some basic extrapolation could be done to take into account the flow of all pairs of input frames and not just the last one. Overall, the approach is not compared to very challenging baselines.

  - I disagree with the answer that the authors gave to a reviewer's question. Denote ground-truth frames as {X_0, X_1 ...} and predicted frames as {Y_1, Y_2, ...}. When asked if the videos at

[Official Review · AnonReviewer3 · rating 6 · confidence 4 · 18 Dec 2016]
**Reasonable paper, can be improved.**

Paper Summary
This paper makes two contributions -
(1) A model for next step prediction, where the inputs and outputs are in the
space of affine transforms between adjacent frames.
(2) An evaluation method in which the quality of the generated data is assessed
by measuring the reduction in performance of another model (such as a
classifier) when tested on the generated data.

The authors show that according to this metric, the proposed model works better
than other baseline models (including the recent work of Mathieu et al. which
uses adversarial training).

Strengths
- This paper attempts to solve a major problem in unsupervised learning
  with videos, which is evaluating them.
- The results show that using MSE in transform space does prevent the blurring
  problem to a large extent (which is one of the main aims of this paper).
- The results show that the generated data reduces the performance of the C3D
  model on UCF-101 to a much less extent than other baselines.
- The paper validates the assumption that videos can be approximated to quite a
  few time steps by a sequence of affine transforms starting from an initial
frame.

Weaknesses
- The proposed metric makes sense only if we truly just care about the performance
  of a particular classifier on a given task. This significantly narrows the
scope of applicability of this metric because arguably, one the important
reasons for doing unsupervised learning is to come up a representation that is
widely applicable across a variety of tasks. The proposed metric would not help
evaluate generative models designed to achieve this objective.

- It is possible that one of the generative models being compared will interact
  with the idiosyncrasies of the chosen classifier in unintended ways.
Therefore, it would be hard to draw strong conclusions about the relative
merits of generative models from the results of such experiments. One way to
ameliorate this would be to use several different classifiers (C3D,
dual-stream network, other state-of-the-art methods) and show that the ranking
of different generative models is consistent across the choice of classifier.
Adding such experiments would help increase certainty in the conclusions drawn
in this paper.

- Using only 4 or 8 input frames sampled at 25fps seems like very little context
  if we really expect the model to extrapolate the kind of motion seen in
UCF-101. The idea of working in the space of affine transforms would be much
more appealing if the model can be shown to really generated non-trivial motion
patterns. Currently, the motion patterns seem to be almost linear
extrapolations.

- The model that predicts motion does not have access to content at all. It only
  gets access to previous motion. It seems that this might be a disadvantage
because the motion predictor cannot use any cues like object boundaries, or
decide what to do when two motion fields collide (it is probably easier to argue
about occlusions in content space).

Quality/Clarity
The paper is clearly written and easy to follow. The assumptions are clearly
specified and validated. Experimental details seem adequate.

Originality
The idea of generating videos by predicting motion has been used previously.
Several recent papers also use this idea. However the exact implementation in
this paper is new. The proposed evaluation protocol is novel.

Significance
The proposed evaluation method is an interesting alternative, especially if it
is extended to include multiple classifiers representative of different
state-of-the-art approaches. Given how hard it is to evaluate generative models
of videos, this paper could help start an effort to standardize on a benchmark
set.

Minor comments and suggestions

(1) In the caption for Table 1: ``Each column shows the accuracy on the test set
when taking a different number of input frames as input" - ``input" here refers
to the input to the classifier (Output of the next step prediction model). However
in the next sentence ``Our approach maps 16 \times 16 patches into 8 \times 8
with stride 4, and it takes 4 frames at the input" - here ``input" refers to
the input to the next step prediction model. It might be a good idea to rephrase
these sentences to make the distinction clear.

(2) In order to better understand the space of affine transform
parameters, it might help to include a histogram of these parameters in the
paper. This can help us see at a glance, what is the typical range of these
6 parameters, should we expect a lot of outliers, etc.

(3) In order to compare transforms A and B, instead of ||A - B||^2, one
could consider A^{-1}B being close to identity as the metric. Did the authors
try this ?

(4) "The performance of the classifier on ground truth data is an upper bound on
the performance of any generative model." This is not *strictly* true. It is
possible (though highly unlikely) that a generative model might make the data
look cleaner, sharper, or highlight some aspect of it which could improve the
performance of the classifier (even compared to ground truth). This is
especially true if the the generative model had access to the classifier, it
could then see what makes the classifier fire and highlight those discriminative
features in the generated output.

Overall
This paper proposes future prediction in affine transform space. This does
reduce blurriness and makes the videos look relatively realistic (at least to the
C3D classifier). However, the paper can be improved by showing that the model can
predict more non-trivial motion flows and the experiments can be strengthened by
adding more classifiers besides than C3D.

[Official Review · AnonReviewer2 · rating 5 · confidence 3 · 19 Dec 2016]
**reasonable paper but the contribution right now is very incremental compared to previous works**

The paper proposes a method for future frame prediction based on transformation of previous frame rather than direct pixel prediction.

Many previous works have proposed similar methods. The authors in their responses state that previous work is deterministic, yet the proposed model also does not handle multimodality.

Further, i asked if they could test their method using 2 RGB frames as input and predicting the transformation as output, to be able to quantify the importance of using transformations both as input and output, since this is the first work that uses transformations as input also. The authors dismissed the suggestion by saying "if we were to use RGB frames as input and ask the model to output future frames it would produce very blurry results", that is, misunderstanding what the suggestion was. So, currently, it does not seem to be a valid novel contribution in this work compared to previous works.

[Final Decision · Program Chairs · 06 Feb 2017]
**ICLR committee final decision**

There were serious concerns raised about the originality of the work in regard to prior methods. The authors' responses to numerous reviewer questions on this matter were also unsatisfactory. In the end, it is difficult to tell what the contribution in regard to the video prediction model is. The proposed evaluation metric is interesting, but also raised serious concerns. Finally, several reviewers raised concerns about the quality of the results.